# The temperature of emotions

**Francisco Barbosa Escobar** [1]*, **Carlos Velasco**[2], **Kosuke Motoki**[3], **Derek Victor Byrne**[1], **Qian Janice Wang**[1]

**1** Department of Food Science, Faculty of Science and Technology, Aarhus University, Aarhus, Denmark,
**2** Department of Marketing, Centre for Multisensory Marketing, BI Norwegian Business School, Oslo, Norway,
**3** Department of Food Science and Business, Miyagi University, Sendai, Japan

* francisco.barbosa@food.au.dk

**Data Availability Statement:** All data and scripts associated with the manuscript are available in OSF. DOI: 10.17605/OSF.IO/TCBG5 URL: https://osf.io/tcbg5/.

**Funding:** This study was partially supported by a Carlsberg Foundation Young Researcher

## Abstract

Emotions and temperature are closely related through embodied processes, and people seem to associate temperature concepts with emotions. While this relationship is often evidenced by everyday language (e.g., cold and warm feelings), what remains missing to date is a systematic study that holistically analyzes how and why people associate specific temperatures with emotions. The present research aimed to investigate the associations between temperature concepts and emotion adjectives on both explicit and implicit levels. In Experiment 1, we evaluated explicit associations between twelve pairs of emotion adjectives derived from the circumplex model of affect, and five different temperature concepts ranging from 0˚C to 40˚C, based on responses from 403 native speakers of four different languages (English, Spanish, Japanese, Chinese). The results of Experiment 1 revealed that, across languages, the temperatures were associated with different regions of the circumplex model. The 0˚C and 10˚C were associated with negative-valanced, low-arousal emotions, while 20˚C was associated with positive-valanced, low-to-medium-arousal emotions. Moreover, 30˚C was associated with positive-valanced, high-arousal emotions; and 40˚C was associated with high-arousal and either positive- or negative-valanced emotions. In Experiment 2 ($N = 102$), we explored whether these temperature-emotion associations were also present at the implicit level, by conducting Implicit Association Tests (IATs) with temperature words (*cold* and *hot*) and opposing pairs of emotional adjectives for each dimension of valence (*Unhappy/Dissatisfied* vs. *Happy/Satisfied*) and arousal (*Passive/Quiet* vs. *Active/Alert*) on native English speakers. The results of Experiment 2 revealed that participants held implicit associations between the word hot and positive-valanced and high-arousal emotions. Additionally, the word cold was associated with negative-valanced and low-arousal emotions. These findings provide evidence for the existence of temperature-emotion associations at both explicit and implicit levels across languages.

## Introduction

People tend to associate emotions with feelings of warmth and coldness as evidenced by colloquial language [1–4]. For example, expressions such as 'warm love', 'chilled with fear', and

Fellowship (CF19-0587) awarded to QJW. The funders had no role in study design, data collection and analysis, decision to publish, or preparation of the manuscript.

**Competing interests:** The authors have declared that no competing interests exist.

'heartwarming' are commonly used in everyday English language. Such types of expressions are also found in other languages. For instance, the phrase 'abrazo cálido', which translates to 'warm hug' is commonly used in the Spanish language. The phrase '温かい心', which means 'warm heart' is commonly used in the Japanese language. Waggoner [5] found that both children and adults describe love and anger with high temperatures, but they associate fear with low temperatures. Furthermore, the concept of emotional temperature is widely used in business and advertising [6]. Aaker et al. [7] defined emotional warmth as a "positive, mild, volatile emotion involving physiological arousal and precipitated by experiencing directly or vicariously a love, family, or friendship relationship" (p. 377). Warmer temperatures, object-based or environmental, can increase feelings of emotional attachment towards friends and family [8].

Previous research has investigated effects of ambient temperature on emotions and behaviors, which highlights the importance of studying the relationship between temperature and emotions. Different studies have found that extreme hot temperatures can negatively affect emotions and wellbeing. For instance, using data from the USA during the period 2008–2013, Noelke et al. [9] found that, compared to average daily temperatures of 10–16˚C, temperatures above 21˚C—especially above 32˚C—significantly reduce positive emotions and increase negative ones. Examining data from 55 countries, Van de Vliert et al. [10] found that people in low-income countries are less happy than the average of the 55 nations when temperatures deviate upwards or downwards from 24˚C. On the other hand, people in high-income countries are happier when temperatures deviate upwards or downwards from 23˚C. Keller et al. [11] found that higher temperatures improve mood and memory during the spring because of the preceding lower temperatures during the winter, but during the summer, higher temperatures negatively affect mood. However, Parker and Tavassoli [12] suggested that people in hot climates, compared to cold ones, present higher affective expressiveness, as well as higher sensitivity to emotional stimulation, both positive and negative.

In terms of behavior, there seems to be a relationship between extreme high temperatures and aggression. Through a series of five experiments, Anderson et al. [13] found that people hold general beliefs that hot temperatures, relative to warmth comfortable ones, increase feelings of anger and hostility, as well as aggressive and violence behaviors. Moreover, they found that temperatures deviating upwards and downwards from 24˚C, increase aggressive behaviors. These results are echoed by Younan et al.'s [14] findings that long-term exposure to high ambient temperatures is associated with aggressive behaviors in children and adolescents in urban areas. Similarly, using data from South Wales, Australia, Stevens et al. [15] found a similar effect of high temperatures on assaults. However, the authors found that high temperatures were associated with a decrease in anger expressed in social media, as measured by counts of angry tweets. The authors suggested that these opposing patterns may occur because higher temperatures promote behaviors that foster assaults (e.g., more time spent outside, social interactions, higher alcohol consumption) that at the same time decrease opportunity, motivation, or ability to be active in social media. Furthermore, related to behaviors, Lee and Sohn [16] analyzed hourly weather data in New York City and yellow taxicab data in 2013 and found a significant positive relationship between extreme cold and hot outdoor temperatures and increased tipping. The authors suggested that this behavior was mediated by mood, as the comfortable indoor temperatures provided by the taxi drivers improved mood and therefore empathy and gratitude, and ultimately generosity, as measured by the tipping amounts.

Regarding product valuation, Zwebner et al. [17] found that exposure to physical and ambient warmth (in the form of thermoclimate, room temperature, and direct contact with a therapeutic pad) can increase product valuation and willingness to pay. The authors found that

these effects were mediated by the positive affective reactions that physical warmth induced [17]. However, Motoki et al. [18] failed to replicate this study.

The associations between emotions and temperature also work inversely, so that emotional states affect temperature assessments. Wang and Liu [19] found that emotional states can influence thermal comfort and thermal sensation during light physical activities. In their study, the authors found that participants' thermal comfort in light activities (i.e., sitting and standing) was lower under the induced boring emotion condition compared to the joyful and the neutral ones. In other words, participants felt warmer under the boring emotion state.

Related to the present study, Bergman et al. [20] studied implicit semantic associations between temperature and emotional valence, as well as implicit associations between the experience of physical temperature and emotional valence using two separate Implicit Association Tests (IATs). The authors found that thermal words related to high temperatures, as well as physical warmth, were associated to positively valanced words. Thermal words related to low temperatures, as well as physical coldness, were related to negatively valanced words.

Despite the documented relationship between temperature and emotions, systematic research on the association between different emotions and physical temperatures (and temperature concepts) is scarce. The present article contributes to the literature on emotions and to the understanding of associations of abstract concepts with concrete experiences by adopting a comprehensive approach in terms of the nature of these associations (explicit and implicit), coverage of emotion adjectives (and their underlying dimensions) and consistency across languages. We aimed to uncover explicit and implicit associations between different temperature concepts and emotion adjectives, and across four different languages, to maximize the generalizability of our results. To this end, we conducted two experiments. In Experiment 1, we used a questionnaire-based design to evaluate people's explicit associations of twelve pairs of emotion words with five different visual temperature scales going from cold (0°C) to hot (40°C) in 10-degree intervals. We conducted Experiment 1 with native speakers of four languages (i.e., English, Spanish, Japanese, Chinese) and explored the consistency in the associations across the languages. In Experiment 2, we conducted Implicit Association Tests (IATs) with two temperature words (*cold* and *hot*) and opposing pairs of emotion adjectives in terms of valence and arousal. The study of associations between temperature and emotions is of great relevance for both the academic and practitioner-oriented literatures due to the documented effect of temperature on emotional states, decision-making, and consumer behavior. Beyond theoretical contributions to the literature on embodied cognition and learning from statistical regularities in the environment, insights from the study of these associations can guide the development of working spaces, products, and marketing activities.

## The origin of temperature-emotion associations

Metaphors are pervasive in people's lives and are present in literature, as well as everyday language. In their early conceptual metaphor theory (CMT), Lakoff and Johnson [21] suggested that the human conceptual system plays an important role in defining reality, and that this conceptual system, is to a large extent metaphorical. Conceptual metaphors are a way of understanding—often abstract—realms of experiences in terms of another—typically concrete—domain. The authors suggested that metaphors unite reason and imagination, and are therefore critical in contributing to our understanding of the world (see also [22, 23]). Hence, metaphors shape the way people think and experience the world. Associations between temperature and emotions may be an illustration of conceptual metaphors in play where one domain is mapped to another one [22]. While people commonly use these temperature-emotion associations to make sense of the world, their origin is not completely understood.

Nonetheless, the theory of constructed emotion provides a useful framework to explain these associations.

The theory of constructed emotion [24, 25] suggests that emotions are categorizations of past experiences molded by embodied knowledge. Emotions do not come from specific brain structures, rather the brain creates them as concepts by parsing situated sensory information, affect, and physiological changes. Based on previous knowledge and experiences, the brain makes predictions and develops situated conceptualizations [26] to identify incoming sensory inputs, find an explanation, and guide responses towards them. For instance, following this constructionist approach, the emotion concepts related to anger found in previous literature may arise from the parsing of the negative valence and high arousal caused by the repeated exposure of extreme temperatures into this emotion category. An important part of the theory of constructed emotion related to the present study is that new sensory information is encoded and unified whenever the brain predicts it will trigger physiological changes in the perceiver [25], and emotion concepts are triggered when the brain infers the meaning of these physiological processes [27, 28]. Hence, emotions are tightly linked to these processes that help in achieving homeostasis [5, 7]. As the theory of constructed emotion would suggest, individuals create emotion categories based on their experience, and these categories can be used to label sensations that occur together with different temperatures [25, 29]. In this way, the associations between temperature and emotions can be explained by the categorization of experiences that coincide and incorporate either bodily temperature feelings or externally perceived temperature, whether the latter relates to the temperature of a living being or an object in close proximity, or whether it relates to environmental temperature.

Bridging the theory of constructed emotion and homeostatic processes can help explain associations between emotions and temperature since both work together to help in maintaining the body's optimal temperature. Emotions can serve as homeostatic responses through two mechanisms [30]. First, emotions can have a direct effect on the body's physiological condition—including temperature changes, and homeostatic processes can bring the body back to thermal equilibrium. For example, being accompanied by a loved person can increase the heartrate and raise the internal body temperature. This often results in blushing, which serves as a thermoregulatory mechanism to cool down the body by drawing more blood to the cheeks, where heat exchange is more efficient [31–33]. In this way, the (increased) internal body temperature experienced would be associated with the situation, as well as its valence and arousal, that triggered the physiological process. The emotion-temperature association can also come from the perception of the warmth generated by the bodily heat of the other person and the core affect and feelings experienced during the encounter. Additionally, emotions can facilitate homeostatic balance by simulating bodily states through thoughts and memories. Bruno et al. [1] found that consumers' perception of emotional stimuli depends on their homeostatic state. For instance, when consumers felt physically cold, they perceived emotionally warm stimuli (i.e., print advertisements that evoked emotions related to love and affection) more favorably than emotionally cold stimuli (i.e., print advertisements that evoked emotions related to disgust). On the other hand, when consumers felt physically hot, they perceived the emotionally cold stimuli more favorably than the emotionally warm ones. However, there were no differences in the evaluation of these stimuli when consumers were at their homeostatic optimum. The authors suggested that these effects occurred because the emotional stimuli brought people further away from their homeostatic optimum when the emotional temperature and the divergence from the optimum was in the same direction.

## Language differences in temperature-emotion associations

It is possible that the associations between temperature and emotions are shaped by language. Previous research has uncovered variations in the subjective experience of emotions and their taxonomy brought by culture and language [34]. Language plays a critical role in cognition because it serves as a medium to ground abstract and general concepts, as well as their emotional connections, in the brain. In this way, emotions have meanings and are experienced differently based on the concepts with which they are associated [35]. Language influences how emotions are conceptualized and categorized across cultures, whether it is through the impact of language on the conceptualization of experiences or the effect of culture on the experience and linguistic representation of emotions [36, 37]. Relevant to the present study, cultural and linguistic differences in emotions may influence the associations between emotions and temperature.

Previous literature has shown differences of emotional and conceptual content across languages [35, 38–40]. Differences in temperature-emotion associations might occur because the underlying meanings and concepts associated to specific emotions can vary with language. There is evidence of significant variations in the associations of emotion concepts in twenty of the language families [27]. Therefore, people across different languages would be evaluating inherently different concepts. Nonetheless, some studies point to a large degree of similarity in the experience of emotions across cultures [41, 42]. Thus, to increase the generalizability of our study and explore the extent to which specific temperature-emotion associations hold beyond one language, we conducted Experiment 1 using four different languages (i.e., English, Spanish, Japanese, Chinese). Furthermore, we analyzed how consistent were the associations across the four languages. We focused on native speakers on these languages since a great part of emotional concepts and understanding is formed interdependently with language during childhood with interactions with caregivers and peers [43–45].

Another potential source of variation in the associations studied here relates to how people experience and evaluate different temperature conditions across cultures, and to how they impact emotions. While warm pleasant temperatures have been associated with positive mood in western countries [11, 46], the way people react to and the emotions that these temperatures elicit, or with which they are associated, may differ due to repeated exposure and the difference in ranges and fluctuations of temperature [47, 48]. An example of the influence of environmental factors on associations with emotions was described by Jonauskaite et al. [49] in a study across fifty-five countries. The authors found that associations between yellow and joyful emotions are stronger when temperatures are moderate, and rain is plentiful. Moreover, related to the role of culture and environmental attitudes, Knez and Thorsson [50] studied evaluations of a park with Japanese and Swedish participants. Japanese participants evaluated the park as warmer and more pleasant than the Swedish participants, despite both groups physically experiencing similar thermal conditions.

## Hypotheses development

We built our hypotheses based on the theory of grounded cognition [51] and the model of core affect [52]. The theory of grounded cognition proposes that abstract concepts are grounded in concrete experiences [51, 53]. A critical aspect of this theory is based on the role of mental simulations on cognition [54], which suggests that the brain captures, merges, and stores different states across modalities—including perception and introspection—during an experience and later re-enacts this cohesive multimodal representation as needed. According to the model of core affect, the cognitive structure of human emotions can be conceptualized and represented by spanning the dimensions of valence and arousal in a circumplex model,

which makes it possible to decompose emotions into varying levels of the two components [52, 55]; see also [56, 57].

In terms of valence, previous research points to a correlation between higher temperatures and positive emotions [1, 17, 18, 20, 58–61]. Nevertheless, the study of associations between emotions and temperature requires a nuanced approach since there are differences in temperature levels (i.e., mild vs. extreme) and how they are experienced (i.e., ambient vs. object-based) that may affect these associations. Some of these studies have used beverages (coffee and tea) as object-based thermal stimuli [59, 60] and have found that beverages at hotter temperatures (above 65˚C) tend to be characterized with more positive emotions. Pramudya and Seo [59] found that when the beverages were at lower temperatures (as low as 5˚C), they elicited more negative emotions.

However, research studying ambient temperature have found that hotter temperatures are associated with negative emotions and well-being. Using Facebook and Twitter data during the period between 2009 and 2016 to analyze the effect of temperature on emotions, Baylis et al., [62] found that expressions of positive emotions peaked at 20˚C. Past 30˚C, positive emotions decreased while negative emotions increased. Noelke et al. [9] found that temperatures above 32˚C significantly reduced positive emotions and increased negative emotions and fatigue in the U.S. population above 18 years old. Other studies have found that higher ambient temperatures tend to increase aggressive behaviors [63–65]. Furthermore, while warm temperatures are correlated with higher mood in the springtime, higher temperatures in the summer are associated with lower mood [11]. Based on this literature, we expected to observe an inverted U-shape relationship between valence and temperature where extreme temperature concepts (both hot and cold) are associated with negative-valanced emotion adjectives while mild warmer temperature concepts are associated with positive-valanced emotion adjectives. More formally, we hypothesized that:

$H_{1A}$: Mildly warm (vs. mildly cool) temperature concepts are associated with positively (vs. negatively) valanced emotion adjectives.

$H_{1B}$: Extreme (vs. mild) temperature concepts are associated with negatively (vs. positively) valanced emotions adjectives.

Regarding the arousal dimension, Cavanaugh et al. [2] found that people tend to differentiate low arousal emotions with the word cold compared to high arousal emotions with the word hot. From a physiological perspective, a few studies have found that higher levels of emotional excitation, independent of valence, can elicit an increase in body temperature. For instance, Salazar-López et al. [66] demonstrated that when participants were presented with high arousal images from the International Affective Picture System (IAPS) with both positive and negative valences, the temperature of the tip of their nose experienced significant increases in temperature. Similarly, Levenson [67] found that increases in skin temperature were correlated with excitement. Hahn et al. [68] found that facial temperature increases in response to arousal during interpersonal interaction. Stress in adults caused by performing complex mental tasks or lying can trigger a temperature increase in the forehead and periorbital regions [69]. Thus, we propose that high-arousal emotions are associated with higher temperatures, whereas low-arousal emotions are associated with colder temperatures. More formally we hypothesize that:

$H_2$: Hotter (vs. colder) temperature concepts are associated with higher (vs. lower) arousal emotion adjectives.

Regarding the effect of language on temperature-emotion associations, given the mixed findings of the literature so far in terms of differences in the experience of emotions across cultures, we aimed to explore whether there was an overall similar pattern in the associations across languages in Experiment 1. Thus, in Experiment 1 we used five levels of temperature concepts and twelve specific emotion adjectives radiating from the valence-arousal core affect model. Furthermore, in Experiment 2, we used opposing adjectives of the core affect model in the two dimensions and two opposing words to refer to temperature.

## Experiment 1

In Experiment 1, we evaluated to what extent the general population associated a range of five specific temperature concepts with twelve different emotion words varying along the dimensions of valence and arousal [70]. In this experiment, we focused on explicit associations and evaluated these potential associations in four languages (i.e., English, Spanish, Japanese, and Chinese).

### Methods

**Participants.** A total of 451 participants completed the experiment online. Participants were recruited through a combination of different methods, including Prolific (https://www.prolific.co/) (*n* = 271), Lancers (https://www.lancers.jp/) (*n* = 61), and the authors' academic and professional network via social media (*n* = 119). Those participants recruited from Prolific received GBP 0.60 for their participation, and those recruited from Lancers received JPY 50. The participants recruited through the authors' academic and professional network consisted of business school students and alumni from different demographic, academic, and professional background. The experiment lasted approximately eight minutes and was performed on Qualtrics (https://www.qualtrics.com/). Data from participants that had an overall duration of two standard deviations below or above the mean were not included in the analysis (*n* = 4). The questionnaire was available in four different languages, namely, English, Spanish, Japanese, and Chinese; and participants were instructed to complete the questionnaire in their first language. To reduce the potential confounding effects of other cultures or languages, only those observation in which participants' reported first language matched the language in which the experiment was completed were retained for the analysis. The final data comprised observations from 403 participants from 28 different countries (221 females, 180 males, and 2 did not reveal their gender; age range = 18–78 years, $M_{age}$ = 35.01 years, $SD_{age}$ = 11.32). The division of participants by language and age statistics is presented in Table 1. The number of participants per country and language is detailed in S1 Table in S1 File.

As a simple online questionnaire, the studies reported in this research comply with the policies and requirements stated by the Aarhus University Research Ethics Committee and was therefore exempted from the need for formal ethical approval. All participants, however, gave their informed written consent before taking part in the experiments.

**Experimental stimuli.** The emotion stimuli consisted of the twelve pairs of adjectives proposed by Jaeger et al. [70] to evenly span the circumplex emotional space defined by valence (x-axis) and arousal (y-axis). S1 Fig in S1 File shows the canonical circumplex model of affect. This set of emotion words allowed us to have high granularity while maintaining a clear guideline on the relevant dimensions to study. Since these emotions are derived from the valence and arousal dimensions, they allowed us to have a better control of the composition of the emotions and reduce potential confounding effects from an endless pool of emotions. Furthermore, this set of emotion words mitigates the risk of vague emotions and poor usability as it

**Table 1. Participant number, age, and starting emotional state of participants by language group in Experiment 1.**

|  | *n* | Age | | Valence | | Arousal | |
|---|---|---|---|---|---|---|---|
|  |  | *M* | *SD* | *M* | *SD* | *M* | *SD* |
| Overall | 403 | 35.01 | 11.32 | 5.94 | 1.54 | 4.49 | 1.81 |
| English | 154 | 33.68 | 12.46 | 6.12 | 1.51 | 6.12 | 1.51 |
| Spanish | 94 | 33.51 | 7.85 | 6.14 | 1.52 | 6.14 | 1.52 |
| Japanese | 79 | 40.92 | 9.50 | 5.29 | 1.44 | 5.29 | 1.44 |
| Chinese | 76 | 33.41 | 12.15 | 5.99 | 1.54 | 5.99 | 1.54 |

has been validated in product-oriented research [70–72] and therefore has the potential to be applied in multiple fields.

**Design and procedure.** Participants consented to take part in the study by agreeing to a statement of informed consent. Before starting the main part of the study, they were asked to evaluate their current emotional state with Bradley and Lang's [73] 9-point Self-Assessment Manikin (SAM) (see Table 1). Following, participants were asked to evaluate how well each of the twelve pairs of emotions matched each of five different temperature concepts (0, 10, 20, 30, and 40 degrees Celsius) with a dropdown box with the options "Not well at all", "Slightly well", "Moderately well", "Very well", and "Extremely well", representing the values 1 to 5 respectively. The different temperatures were presented with a visual representation of an alcohol thermometer filled at different levels with the temperature stated in degrees Celsius (°C) and its equivalent in degrees Fahrenheit (°F) (see Fig 1). These temperatures were chosen since they cover a spectrum from relatively cold to relatively hot, including a standard reference for room temperature level at 20°C [74]. Moreover, the visual representation of the temperatures was chosen to provide a common frame of reference for temperature and to mitigate potential confounds brought by personal and language-related interpretations of different temperature words. The order of the emotion words and the temperature visual representations was randomized for each participant. After evaluating the twelve emotion words, participants were asked some demographic questions (i.e., age, gender, first language, current country of residence, closest city).

**Data analysis.** First, the mean ratings of the associations between temperatures and emotions including all languages were computed. To obtain an overall perspective of the associations between emotion adjectives and temperature concepts, a Principal Component Analysis (PCA) with Varimax rotation was performed on the mean ratings of the association with each

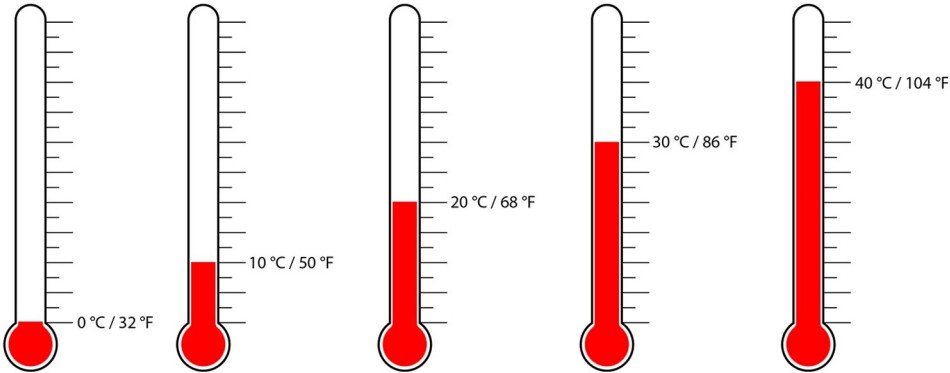

**Fig 1. Temperature visual scales used in Experiment 1.** The scales correspond to 0, 10, 20, 30, and 40 Degrees Celsius (°C) and their equivalent in Degrees Fahrenheit (°F).

temperature for all emotions within each language. In all the plots, the color coding of the emotion words follows the order of Jaeger et al.'s [70] circumplex starting with *Tense/Bothered* in the second quadrant and moving clockwise. The circumplex model in the four languages can be seen in S1 Fig in S1 File.

Furthermore, given that the data violated the assumptions of normality and homoscedasticity, separate rank-based analysis of variance-type statistics (ATS) were conducted on the association ratings for each temperature. Emotion (12 levels) was taken as a within-subject factor and language (4 levels) as a between-subject factor. The ATS were conducted using the R Statistical software package {nparLD} [75]. ATS is suitable for studies with non-normal distributions (as it makes virtually no assumptions regarding the distribution of the data), small sample sizes, and outliers [76]. Moreover, ATS does not assume equal variances as other non-parametric methods, such as the Kruskal-Wallis test or the van der Waerden (VDW) test do. Compared to classical non-parametric statistical methods (e.g., Mann-Whitney-Wilcoxon, Kruskal-Wallis) that are not robust to heteroscedastic data, as well as factorial designs, ATS does not suffer from these limitations [76, 77], and it performs better than the Wald-type statistic as it better controls for the type I error in multiway factorial designs [78]. Next, Bonferroni-Holm corrected Wilcoxon Signed Rank tests were performed on the significant effects of temperature within each language.

In addition, to have a better understanding of the trend in the associations between temperature and the circumplex model of emotions, as well as the overall effect of language, age, and gender, we converted the association ratings of the different temperature categories into a continuous variable and fit a Linear Mixed Model (LMM) using the *lmer* function of the {lme4} R package [79]. To compute the continuous temperature variable, we first transformed the 1–5 rating scale to a 0–4 scale to avoid confounds between ratings of 1 and associations with 0˚C. Next, we calculated the weighted average of the temperature categories. The *p*-values of the LMM were calculated with the Satterwhite method using the {lmerTest} [80] and {pbkrtest} [81] R packages. We incrementally fit three LMM models using the continuous temperature measure as dependent variable and tested them using a likelihood ratio test (LRT). Moreover, the Akaike Information Criterion (AIC) and the Bayesian Information Criterion (BIC) were used to select the best fitting model. The reduced null model (Model 0) included only subject ID as a random factor; Model 1 included the interaction between emotion and language; age and gender were added as covariates in Model 2.

To analyze the overall degree of similarity of the associations across the different languages, Pearson's correlations (*r*) between each pair of language's 12 (emotions) × 5 (temperatures) ratings matrices were computed. This procedure was similar to the one used by Jonauskaite et al. [82] to analyze the similarity patterns in the associations of colors and emotions across languages.

## Results

To determine the overall temperature-emotion associations, a matrix with the average ratings including all the data was created (see Fig 2). The highest mean values for 0˚C were observed with *Blue/Uninspired*, and for 10˚C, the highest values occurred with *Passive/Quiet* and *Blue/Uninspired*. For 20˚C, the highest mean values were with *Secure/at ease*, *Relaxed/Calm*, and *Happy/Satisfied*. For 30˚C the highest mean values occurred with *Energetic/Excited*, and for 40˚C, the highest values occurred with *Energetic/Excited* and with *Tense/Bothered*.

The PCA revealed that the first two dimensions explained 96.53% of the variance in the data, where the first principal component (PC1) accounted for 69.20%, and the second principal component (PC2) accounted for 27.33% (the corresponding factor loadings are presented

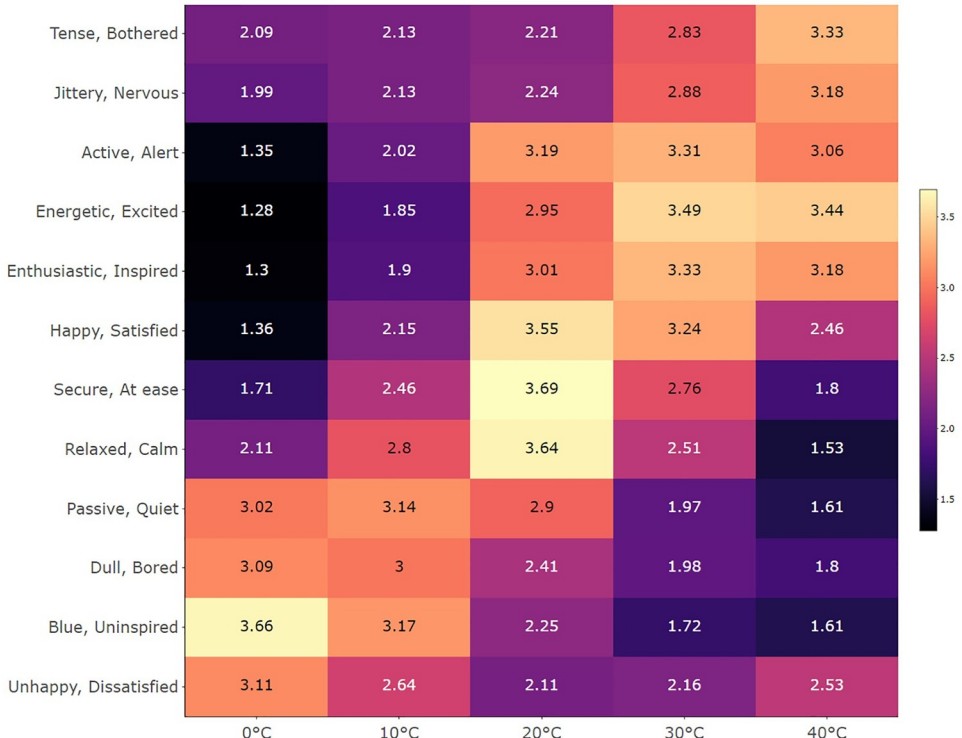

**Fig 2. Heatmap of overall associations in Experiment 1.** The heatmap shows the average ratings of the temperature-emotion associations with all the data. Less saturated red or orange indicates stronger associations.

in S2 Table in S1 File). The PCA revealed that the temperature categories were associated with different areas of the circumplex (Fig 3). Given these results, it seems as if the first and second dimensions described arousal and valence, respectively. In the PCA projection, increasing temperatures moved clockwise starting with 0˚C in the third quadrant and ending with 40˚C in the fourth quadrant.

When the results are interpreted referenced to the "canonical" circumplex model of affect (S1 Fig in S1 File), in which valence is located in the x-axis and valence is located in the y-axis, they revealed a counterclockwise movement in the associations between emotions and temperature. More specifically, the emotions on the third quadrant—including *Blue/Uninspired*—were associated with 0˚C. From here, the associations moved counterclockwise in the circumplex model of core affect with increasing temperatures, ending at 40˚C in the second quadrant, which includes the *Tense/Bothered* emotion.

Moreover, as the PCA showed, the associations between temperature and emotions were highly similar across the four languages.

The ATS analysis revealed significant main effects and interactions for all the temperatures, except for the main effect of language in the 40˚C temperature (see Table 2). For 0˚C, the analysis revealed significant main effects of emotion and language, as well as significant effects of their interaction. For 10˚C, emotion, language, and their interaction were significant. Moreover, for the 20˚C temperature, the analysis revealed significant main effects of emotion, language, and their interaction. For 30˚C, emotion and language, as well as their interaction were significant. Finally, for the 40˚C temperature, the analysis revealed a significant main effect of emotion and a significant interaction effect between emotion and language. However, the analysis did not reveal a significant main effect of language.

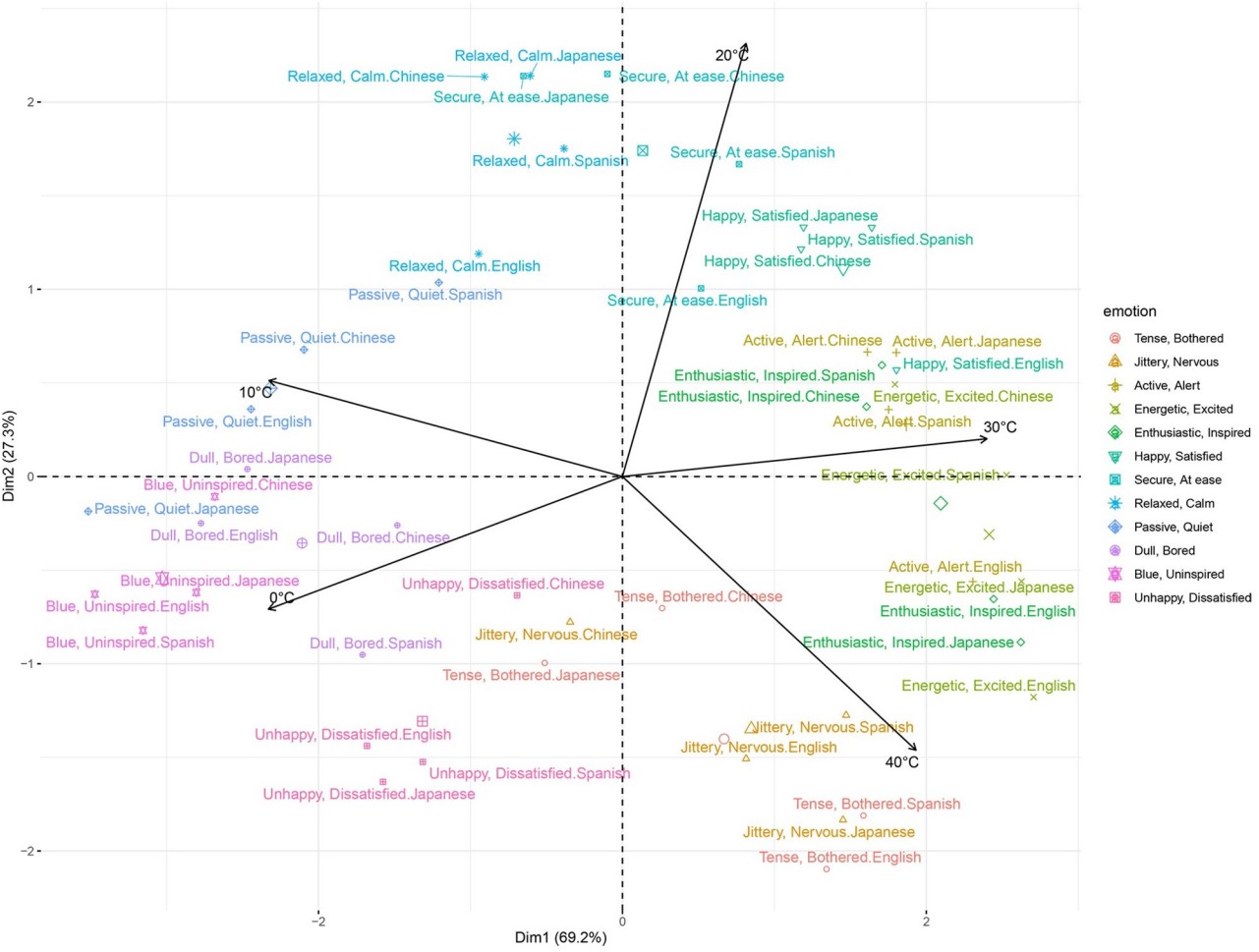

**Fig 3. Principal Component Analysis (PCA) biplot in Experiment 1.** The variables visualized are the temperature-emotion association ratings for the different temperatures. The individual observations of emotions per each language (in different colors and shapes) are superimposed. The emotion categories are color-coded following the canonical circumplex model of affect, starting with *Tense/Bothered* in the third quadrant. Dimension 1 seems to describe arousal, and Dimension 2 seems to describe valence. Associations in the PCA move clockwise with increasing temperature starting from 0˚C in the third quadrant, which translates into a counterclockwise movement in the canonical circumplex model of affect.

Furthermore, the Wilcoxon Signed Ranked test revealed significant differences between various emotion words for each temperature in all the languages. As Fig 4 showed, there were peaks in the average ratings of emotions corresponding to different areas to the circumplex as temperature increases. More specifically, as temperature moved from 0˚C to 40˚C, the highest association ratings moved from the emotions on the rightmost side, next to the middle, and lastly to the leftmost side. These results are consistent with the PCA, and they revealed that the associations moved counterclockwise from the lower left side of the circumplex (third quadrant) to the upper left side (second quadrant). The main and interaction relative treatment effects are presented in S3 Table in S1 File.

The LMM results of Model 1 (with continuous temperature as dependent variable, the interaction of emotion and language as key independent variables, and subject ID as random factor) revealed significant main effects of emotion, $F_{(11,4341)} = 198.48$, $p < .001$, and language, $F_{(3,394)} = 4.43$, $p = .004$, as well as a significant interaction effect of emotion and language, $F_{(33,4341)} = 6.73$, $p < .001$. The results of Model 2 (Model 1 plus age and gender as covariates)

**Table 2. ATS results in Experiment 1.**

| Factor | df | F | p-value |
|---|---|---|---|
| **Temperature: 0˚C** | | | |
| Emotion | 6.874 | 171.32 | < .001 |
| Language | 2.87 | 4.98 | .002 |
| Emotion × Language | 18.07 | 5.31 | < .001 |
| **Temperature: 10˚C** | | | |
| Emotion | 7.547 | 88.28 | < .001 |
| Language | 2.881 | 12.68 | < .001 |
| Emotion × Language | 20.819 | 4.08 | < .001 |
| **Temperature: 20˚C** | | | |
| Emotion | 6.046 | 145.92 | < .001 |
| Language | 2.862 | 14.68 | < .001 |
| Emotion × Language | 16.582 | 3.03 | < .001 |
| **Temperature: 30˚C** | | | |
| Emotion | 7.181 | 143.81 | < .001 |
| Language | 2.857 | 3.41 | .018 |
| Emotion × Language | 19.518 | 3.46 | < .001 |
| **Temperature: 40˚C** | | | |
| Emotion | 5.745 | 122.77 | < .001 |
| Language | 2.627 | 0.78 | .492 |
| Emotion × Language | 15.76 | 5.85 | < .001 |

revealed main effects of emotion, $F_{(11,4341)} = 198.47$, $p < .001$, and language, $F_{(3,394)} = 4.63$, $p = .003$, as well as a significant interaction effect of emotion and language, $F_{(33,4341)} = 6.74$, $p < .001$. However, age, $F_{(1,390)} = 0.54$, $p = .545$, and gender, $F_{(2,423)} = 1.81$, $p = .165$, were not significant. The LRT revealed that Model 1 was significantly better than Model 0 (null model with only subject ID as random factor), $\chi^2_{(47)} = 2208.42$, $p < .001$. Adding age and gender in Model 2 did not contribute towards explaining the data, $\chi^2_{(3)} = 4.10$, $p = .251$. In addition, the model comparison showed that both AIC and BIC decreased from 34781 and 34800, respectively, for Model 0, to 32667 and 32990, respectively, for Model 1. Both AIC and BIC increased to 32668 and 3,012, respectively, for Model 2. We also explored controlling for the recruitment platform (Prolific, Lancers, authors' networks) in this model. Our variables of interest (i.e., emotion, language, and their interaction) remained the same, while gender, $F_{(2,421)} = 3.29$, $p = .038$, and platform, $F_{(2,389)} = 7.21$, $p < .001$, were significant. Although there may be small differences in the temperature ratings based on gender and recruitment platform (main effects), overall, our results are largely unchanged. We therefore proceeded with the most parsimonious Model 1. The LMM analysis provided further evidence that, overall, the associations with emotions as temperature increased moved counterclockwise in the circumplex model of affect (Fig 5). These associations were fairly consistent across languages, with the largest differences occurring in the *Tense/Bothered* and *Jittery/Nervous*. Moreover, age and gender did not have a significant effect in our data.

Moreover, the results of the LMM revealed that, while the temperatures associated with the different emotions were highly consistent across languages, there were some clear differences. In particular, the Chinese-speaking participants seemed to associate high arousal emotions, independent of valence, with lower temperatures compared to the speakers of the other languages. This was more prominent for the *Tense/Bothered* and *Jittery/Nervous* (negative valence emotions); although it was similar to Japanese in *Tense/Bothered*. In addition, the largest

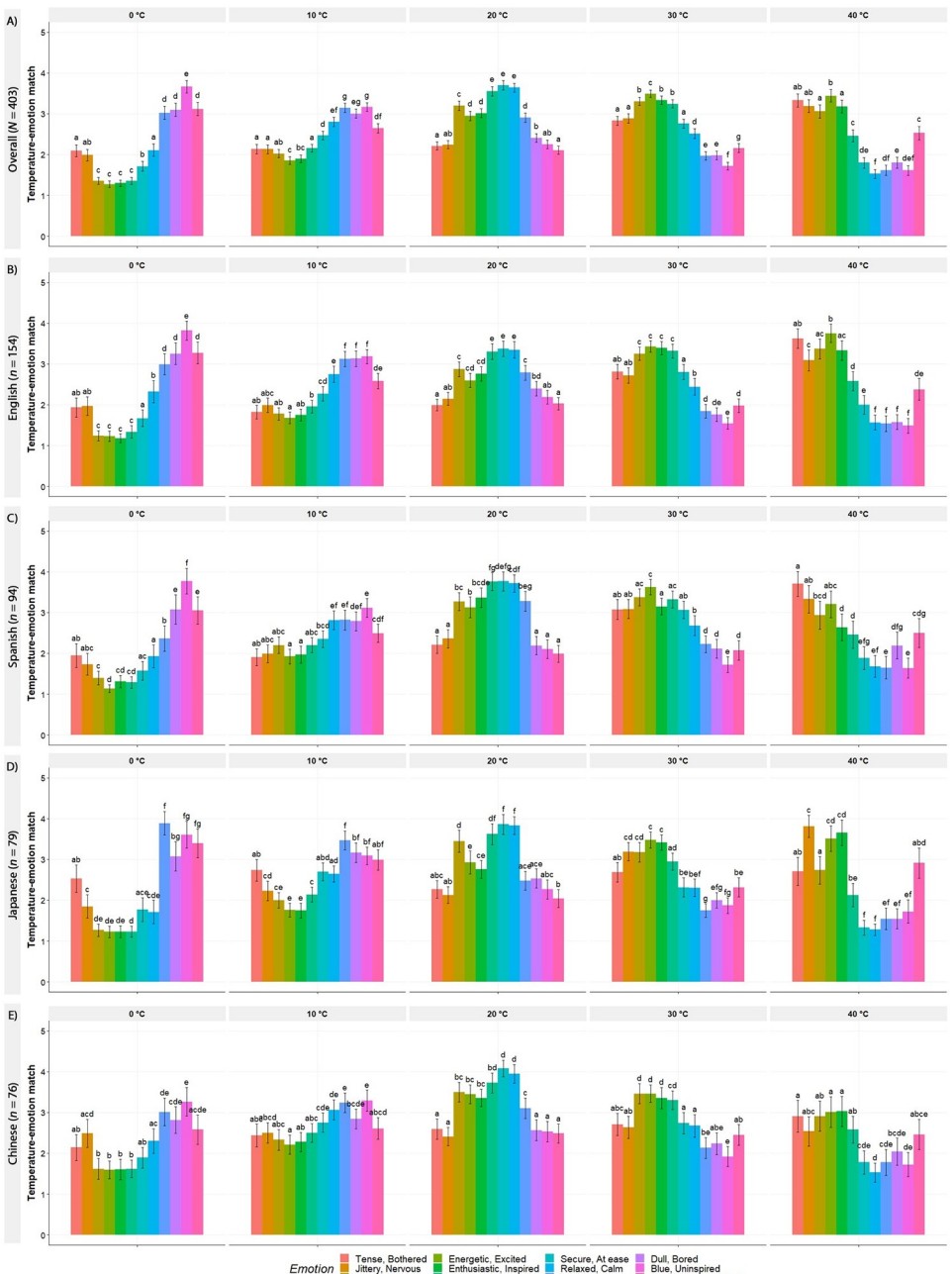

**Fig 4. Mean ratings and Wilcoxon Signed Ranked test results for the temperature-emotion association ratings in Experiment 1.** The y-axis is on a 1–5 scale, where 1 indicates the emotion and the temperature do not match well at all, and 5 indicates they match very well. The plots are divided into different blocks for each temperature (horizontally) within each language (vertically). The error bars represent the standard errors of the mean. The letters represent the different significance groups ($p < .05$) within each temperature and language as per the Wilcoxon Signed Rank Test.

heterogeneity across the languages occurred in the *Enthusiastic/Inspired* and *Passive/Quiet emotions*.

The analysis also revealed large similarities in the associations between emotion adjectives and temperature concepts across languages. All the language-to-language similarity levels ($r$) were above.80. The highest degree of similarity occurred between Spanish and English

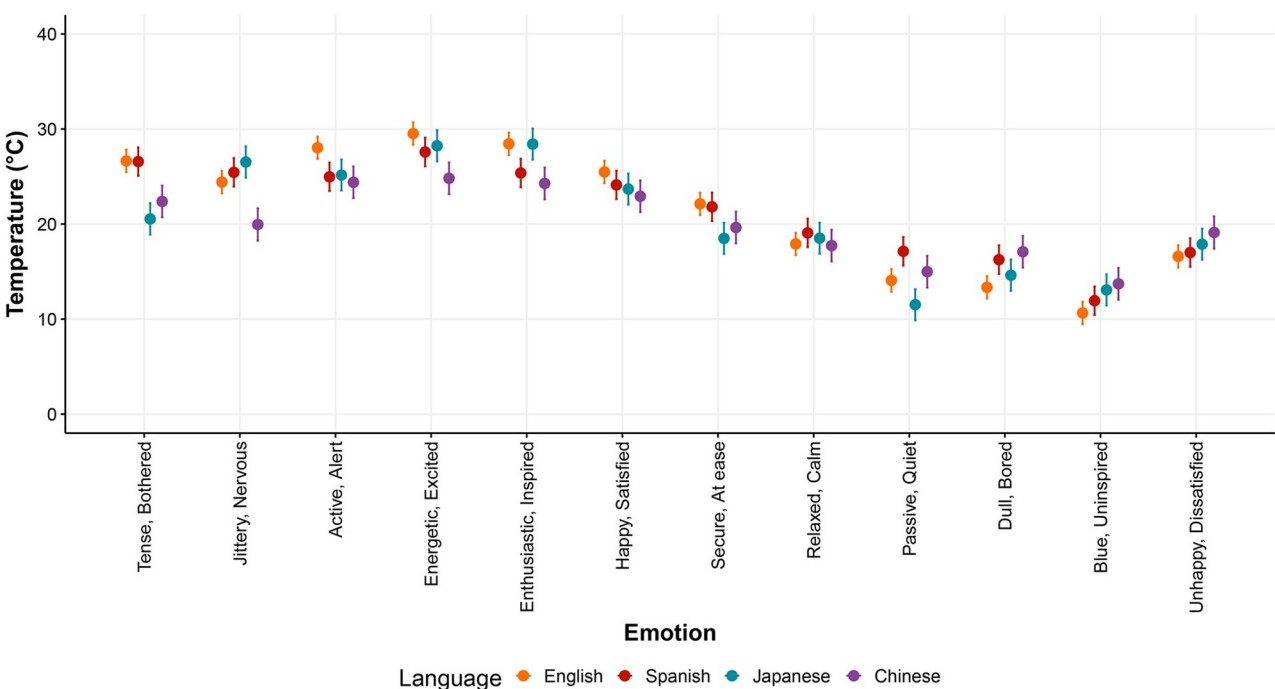

**Fig 5. Mean temperature association ratings following Model 1 of the linear mixed model (LMM) analysis using temperature as a continuous variable.** The emotion adjectives appear in the x-axis, and the languages are color-coded. The error bars represent the confidence intervals.

($r = .920$, 95% CI = [.870, .952], $p < .001$) followed by Chinese and Spanish ($r = .895$, 95% CI = [.829, .936], $p < .001$). The lowest similarity level was between Japanese and Spanish ($r = .837$, 95% CI = [.741, .900], $p < .001$). See Fig 6 for all the language-to-language similarity levels.

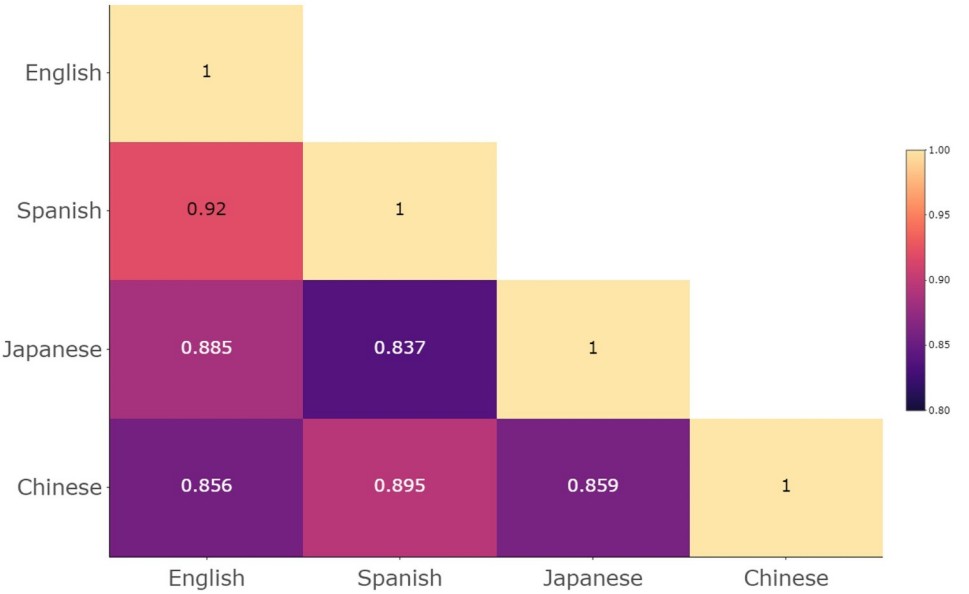

**Fig 6. Language-to-language associations similarity in Experiment 1.** The matrix shows the Pearson correlation values ($r$) between each pair of language's 12 × 5 temperature-emotion associations matrix. Higher values indicate higher similarity between the associations.

Overall, the results of Experiment 1 provided evidence that participants had consistent explicit associations between specific areas of the valence-arousal emotion circumplex and different temperature concepts. The highest mean associations ratings moved from the lower left side of the circumplex to the upper left side as temperature increased. The results revealed that the lower temperatures (0˚C and 10˚C) were associated with negatively valanced, low arousal emotions (third quadrant). Additionally, 20˚C was associated with positively valanced, neutral arousal emotions (positive x-axis), as well as positively valanced, low arousal emotions (fourth quadrant). The warm temperature (30˚C) was associated with positively valanced, high arousal emotions (first quadrant). Finally, 40˚C was associated with high arousal emotions, both positively (first quadrant) and negatively (second quadrant) valanced. These findings revealed that in the mild range, higher temperature concepts were associated with more positively valanced emotion adjectives, which supported $H_{1A}$. The results provided partial support for $H_{1B}$ as the "extreme" cold and hot temperature concepts (0 and 40˚C) were associated with negatively valanced emotion adjectives, but 40˚C was also associated with a positive emotion adjective. Furthermore, the findings showed a linear relationship between higher temperature concepts and high arousal emotion adjectives, supporting $H_2$. Despite some minor differences, the emotions that the participants from the four languages associated with the different temperatures were highly correlated. However, it is important to note that our sample included native speakers of the same specific language from different countries. Hence, due to the specificities of a same language across countries and regions, the sample size does not allow for a generalization of these associations across the languages. Moreover, there may be potential confounding factors, such as country of residence and birth, and dialect, among others affecting the languages.

## Experiment 2

The purpose of Experiment 2 was to expand the findings of Experiment 1 and evaluate whether people also exhibit associations between emotions and temperature concepts at an implicit level. To this end, we conducted a modified Implicit Associated Test (IAT) similar to the one employed by Crisinel and Spence [83] and Chen et al. [84], including the improved IAT scoring algorithm suggested by Greenwald et al. [85]. Given that the emotions evaluated in Experiment 1 were derived from different combinations in the valence-arousal space and these dimensions were related to the temperatures, in Experiment 2, we used the positive and negative poles of both dimensions according to the emotion circumplex proposed by Jaeger et al. [70] as stimuli in two separate versions of the IAT. Furthermore, as the different languages shared highly similar associations, in this experiment, we focused only on English-speaking participants from a single country (UK). Experiment 2 is similar to Bergman et al. [20] where the authors conducted an IAT in which participants were tasked to categorize ten thermal words or ten emotion words. However, in said study, the authors only evaluated valence.

### Methods

**Participants.**   A total of 102 native English-speaking participants from the UK took part in the online experiment. Two participants indicated they were also native speakers of another language in addition to English, one French and the other one Serbian. They were recruited from Prolific (https://www.prolific.co/) and received GBP 0.60 for their participation. The experiment was conducted on Gorilla (https://gorilla.sc/) and took nine minutes on average. Following Greenwald et al.'s [85] improved IAT scoring algorithm, the data from one participant was excluded from the study as 10% of their trials had response times greater than 300 ms. At the beginning of the experiment, participants were automatically and randomly

**Table 3. Blocks and response mappings in Experiment 2.**

| Block | Task | Valence | | Arousal | | Trials |
|---|---|---|---|---|---|---|
| | | Left key | Right key | Left key | Right key | |
| 1 | Single—Emotion | Unhappy, Dissatisfied | Happy, Satisfied | Passive, Quiet | Active, Alert | 6 |
| 2 | Single—Temperature | Cold | Hot | Cold | Hot | 6 |
| 3 | Combined—Congruent | Unhappy, Dissatisfied / Cold | Happy, Satisfied / Hot | Passive, Quiet / Cold | Active, Alert / Hot | 16 |
| 4 | Combined—Congruent | Unhappy, Dissatisfied / Cold | Happy, Satisfied / Hot | Passive, Quiet / Cold | Active, Alert / Hot | 48 |
| 5 | Single—Emotion | Happy, Satisfied | Unhappy, Dissatisfied | Active, Alert | Passive, Quiet | 6 |
| 6 | Combined—Incongruent | Happy, Satisfied / Cold | Unhappy, Dissatisfied / Hot | Active, Alert / Cold | Passive, Quiet / Hot | 32 |
| 7 | Combined—Incongruent | Happy, Satisfied / Cold | Unhappy, Dissatisfied / Hot | Active, Alert / Cold | Passive, Quiet / Hot | 48 |

assigned to a version of the IAT test involving emotion words along either the valence (Experiment 2A) or the arousal (Experiment 2B) dimension. The final dataset comprised 51 participants (32 females and 19 males; age range = 18–74 years, $M_{age}$ = 31.27 years, $SD_{age}$ = 12.84) who completed the valence dimension, and 50 participants (34 females and 16 males; age range = 19–73 years, $M_{age}$ = 35.08 years, $SD_{age}$ = 14.24) who completed the arousal dimension.

**Design and procedure.** The IAT was programmed and conducted in Gorilla, and participants completed the experiment online using their desktop or laptop computers. The experiment followed a similar design to the one used in Crisinel and Spence [83]. The experiment comprised seven blocks (see Table 3) in which participants were asked to match the stimulus according to a specific mapping by pressing one of two keys (i.e., left key "F" and right key "J"). The first two blocks consisted of the single task practice trials in which participants matched an emotion word (first block) and a temperature word (second block) according to the corresponding key assigned to it. The third and fourth blocks consisted of the combined task experimental trials where participants matched either an emotion word or a temperature word according to the given mapping. The fifth block consisted of the reversed emotion single task practice trials. The sixth and seventh blocks consisted of the reversed combined task experimental trials.

In the combined tasks, responses were mapped based on the findings of Experiment 1. The hot and the positive valence/high arousal words were mapped to the same key, which constituted the congruent blocks. On the same token, the cold and negative valence/low arousal words were mapped to the same key, which constituted the incongruent blocks (see Fig 7). The order in which participants were exposed to either the congruent trials first, in blocks three and four or later in blocks sixth and seventh, was randomized. Moreover, the response keys assigned to the temperature words were counterbalanced (i.e., left key to the word cold or the word hot). According to Greenwald et al.'s [85] suggestion, all the combined task trials were used in the analysis.

Participants first agreed to the informed consent and were then given general instructions for the experiment. Participants were instructed to maintain their fixation on the center of the screen and to match the stimulus presented as promptly and accurately as possible according to a given mapping by pressing either the "F" (left) or the "J" (right) key. The mappings were provided to participants before each block of trials. Moreover, during the trials, the words assigned to each key remained on the upper left- and right-hand sides of the screen, respectively. Each trial started with a fixation cross at the center of the screen for a randomized interval of 500–600 ms. Later, an interstimulus interval of 300–400 ms was presented. Next, the target stimulus (either a temperature or an emotion word) was presented. If the incorrect key was pressed, a red "X" appeared in the center of the screen. Participants needed to press the

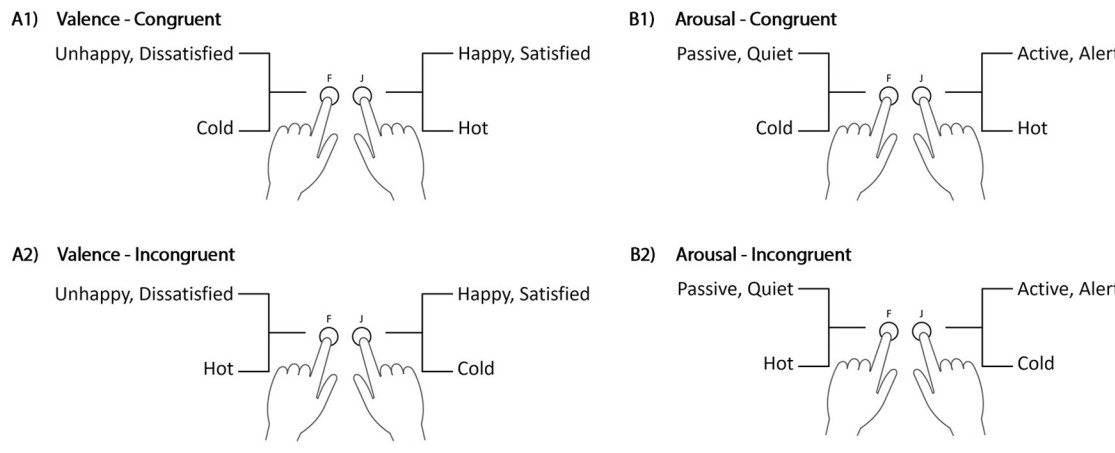

**Fig 7. Stimulus-response key assignment in Experiment 2.** The left-hand side (A1, A2) corresponds to assignments in the valence dimension, and the right-hand side (B1, B2) corresponds to the arousal dimension. Congruent assignments are presented in the upper panels and incongruent assignments are presented in the lower panels.

correct key to continue. Given that the experiment was conducted online, different factors, such as device and internet connection speeds, could affect the timings in the study. The intervals in which the target stimulus was presented were of special importance since participants could miss the stimuli due to low technical capabilities. Thus, to avoid participants missing the stimuli, as well as large discrepancies in the timing, the target stimuli remained visible until the correct key was pressed. That being said, Gorilla has been proven to be an accurate and precise online experiments platform [86–88] and a top performer compared to the most popular packages, including PsychoPy. Bridges et al. [87] conducted a large study investigating the accuracy and precision of visual and auditory stimulus timing and response times of several experiment platforms. The authors found that with visual stimuli, Gorilla had low variability across browsers (Firefox, Chrome, Safari, Edge) and operating systems (Windows, Ubuntu, Macintosh). It presented under 6 ms inter-trial variability in all browsers and sub-millisecond variability in Chrome and Ubuntu. Hence, the performance of the platform provides great confidence in the results.

In the single task practice trials, each stimulus was repeated 3 times. Thus, blocks 1, 2, and 5 consisted of 6 trials each. In the combined task experimental trials, each stimulus was repeated four times in block 3, eight times in block 6, and twelve times in blocks 4 and 7. Hence, block 3 consisted of 16 trials, block 6 consisted of 32 trials, and blocks 4 and 7 consisted of 48 trials each. Therefore, participants completed a total of 144 experimental trials (Table 3 provides further details on the blocks and trials).

**Data analysis.** First, ATS were conducted on the Response Times (RTs) for the congruent and incongruent conditions. Bonferroni-Holm corrected Wilcoxon Signed Rank tests were performed on the significant main effects. To measure effect sizes, Cliff's Delta (CD) was used as per the R package {effsize} [89]. CD values range from -1 to 1. A value of 0 means total overlap between the distributions, while values of -1 and 1 mean no overlap [90]. Next, D values using RTs were analyzed following the improved scoring algorithm proposed by Greenwald et al. [85] (see also [83]). More specifically, the response times of the error trials were replaced by the mean response time of the block plus two standard deviations. This treatment allows the inclusion of error rates while introducing a penalty. As Richetin et al. [91] found, error rates should not be discarded since they contain important information that increases the validity and reliability of the analysis, in addition to increase the statistical power. The RTs of

the correct trials, along with the modified response times of the incorrect trials were used to compute D values. The D values consisted of the mean of the difference in RTs between the initial congruent and incongruent blocks and between the last congruent and incongruent blocks, each divided by their pooled standard deviation. The D values were analyzed with one-sample t-tests. Finally, error rates for the congruent and incongruent conditions were analyzed similarly to the response times.

## Results

The ATS revealed a significant effect of congruence on response times in the valence dimension, $F_{ATS}(1, \infty) = 47.23$, $p < .001$, as well as the arousal dimension, $F_{ATS}(1, \infty) = 46.37$, $p < .001$. The results showed significant differences between the conditions (see Fig 8A). Regarding the valence dimension, participants responded significantly more rapidly when the negative valence emotion was paired with the word cold and the positive valence emotion with the word hot ($M = 684.36$ ms, $SD = 188.55$) that when the negative valence emotion was paired with the word hot and the positive valence emotion was paired with the word cold ($M = 821.24$ ms, $SD = 243.09$; $p < .001$, CD = .408, 95% CI [.189, .588]). In the arousal dimension, participants responded more rapidly when the deactivation emotion was paired with the word cold and the activation emotion was paired with the word hot ($M = 626.96$ ms, $SD = 178.13$) than when the deactivation emotion was paired with the word hot and the activation emotion with the word cold ($M = 924.67$ ms, $SD = 1045.97$; $p < .001$, CD = .478, 95% CI [.264, .648]).

Furthermore, the one-sample $t$-test of the D values was significantly different than zero for both the valence ($M = 0.51$, $SD = 0.49$, $t_{50} = 7.46$, $p < .001$) and the arousal ($M = 0.57$, $SD = 0.53$, $t_{49} = 7.62$, $p < .001$) versions of the IAT tests. These results indicated that, in the valence dimension, the congruent associations—between the negative valence emotion (*Unhappy/Dissatisfied*) and the word cold, and between the positive valence emotion (*Happy/Satisfied*) and the word hot—were stronger than the incongruent associations—those between the negative valence emotion and the word hot, and between the positive valence emotion and the word cold. Moreover, in the arousal dimension, the congruent associations between the deactivation emotion (*Passive/Quiet*) and cold, and between the activation emotion (*Active/*

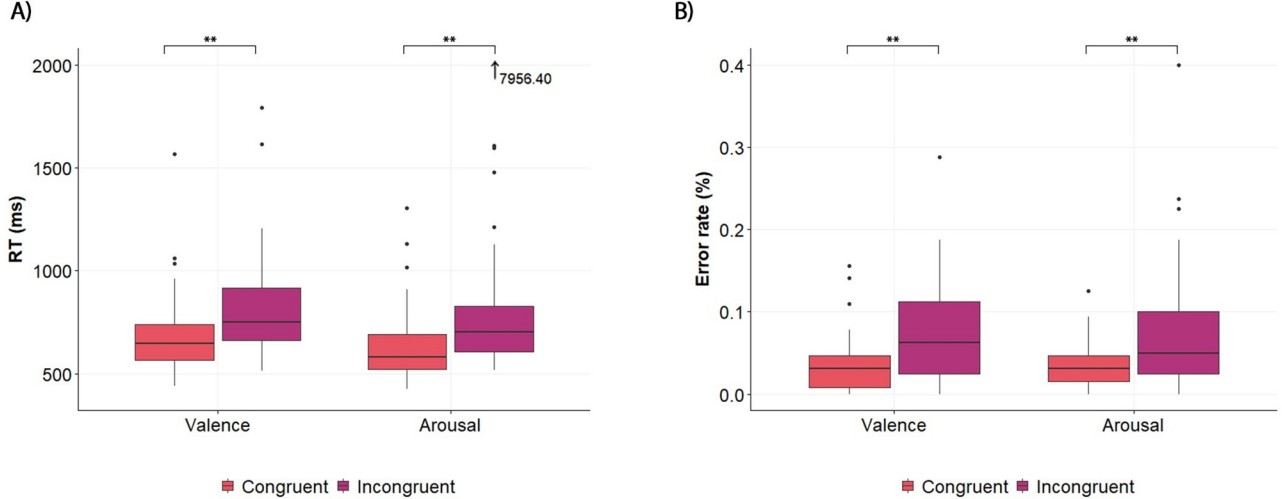

**Fig 8. Experiment 2 results.** Mean response times (A) and mean error rates (B) in the congruent pairings and incongruent pairings in the valence and arousal dimensions. Error bars represent the standard errors of the mean. The asterisks indicate statistically significant differences at $p < .01$ (**).

*Alert*) and hot, were stronger than the incongruent associations between the deactivation emotion and hot, and between activation and cold. To obtain a sense of the magnitude of these effects, the effect sizes found here were stronger those reported in studies using similar methodologies. For instance, taking an example of a study in crossmodal correspondences, Crisinel and Spence [83] reported an effect size of D = 0.29 for the association between sweet tastes and high-pitched sounds.

The analysis of the error rates showed a similar pattern. There was a significant main effect of congruence in the valence dimension, $F_{ATS}(1, \infty) = 28.46$, $p < .001$, as well as the arousal dimension, $F_{ATS}(1, \infty) = 11.05$, $p < .001$. In terms of valence, participants' error rates in the congruent pairings ($M = 0.034$, $SD = 0.035$) were significantly lower than in the incongruent ones ($M = 0.072$, $SD = 0.060$; $p < .001$, CD = .407, 95% CI [.187, .588]). Similarly, in the arousal dimension, participants made significantly fewer mistakes in the congruent pairings ($M = 0.037$, $SD = 0.032$) than in the incongruent ones ($M = 0.074$, $SD = 0.077$; $p < .001$, CD = .269, 95% CI [.036, .474]) (See Fig 8B).

Taken together, the results of Experiment 2 demonstrated that participants exhibited implicit associations between the word hot and words related to positively valanced emotions and high arousal emotions. Moreover, they presented implicit associations between words related to negatively valanced and low arousal emotions with the word cold. In terms of valence, the results are partially consistent with $H_{1B}$ but seem to be more in line with $H_{1A}$ since the word cold, but not hot, was associated with the negative valanced emotion. It is possible that the casual use of the word hot is not necessarily associated with 40°C or extreme temperatures. These results are consistent with those in Bergman et al.'s [20] study where the authors found that participants associated higher temperature words with positive emotions. Regarding arousal, the findings provided support for $H_2$. It is important to note that the difference in response times between the congruent and incongruent matches was slightly weaker for the valence dimension than for arousal, which could indicate a more clear-cut relationship between temperature and arousal, compared to valence. Despite these results, it cannot be discarded that the words hot and cold could have different semantic meanings besides temperature.

## General discussion

In the present study, we aimed to uncover how people associate a range of adjectives spanning the emotional circumplex model with different temperature concepts. To this end, we conducted two experiments. In Experiment 1, we evaluated the explicit associations between twelve different emotion adjectives, varying in valence and arousal, and five different temperature concepts on native speakers of four different languages (English, Spanish, Japanese, and Chinese). In Experiment 2, we evaluated native English speakers in terms of their implicit associations between temperature words (*hot* and *cold*) and emotion adjectives at the opposite ends of both the valence (*Unhappy/Dissatisfied* and *Happy/Satisfied*) and the arousal (*Passive/Quiet* and *Active/Alert*) dimensions.

Altogether, the results provided evidence for the existence of explicit and implicit associations between emotions adjectives and temperature concepts. The results of Experiment 1 showed that, regardless of language, the peak of the association ratings moved counterclockwise in the canonical circumplex model of core affect from the lower left side (third quadrant) to the upper left side (second quadrant) as temperature increased from 0°C to 40°C. The results of the IATs in Experiment 2 revealed that participants had faster response times when the word *hot* was independently matched with the positive-valence and the high-arousal emotion words, than when these emotion words were matched with the word *cold*. Furthermore,

as evidenced by the magnitude of the D values, the associations in the arousal dimension were stronger than in the valence dimension, potentially due to a more linear relationship between temperature and arousal, compared to valance. Therefore, consistent with Experiment 1, the results of Experiment 2 revealed that participants implicitly associated the word cold with the low arousal emotion and the word hot with the high arousal emotion. While some studies have hinted at the existence of temperature-emotion associations, to the best of our knowledge, this is the first study to uncover associations between them explicitly and implicitly and explore their similarity across languages.

Our results may be interpreted from the theory of grounded cognition [51, 53]. It is possible that the associations uncovered here arise from the multimodal representations encoded in the brain incorporating the temperature (body or ambient) experienced during specific emotional states repeated. For instance, the robust associations of 20°C with positive valence, low arousal emotions can be a product of the physical comfort this temperature generates for most people. It is also possible that the associations between emotions and temperature arose because there is causal relationship between them, whether it is in the case where emotions trigger physiological responses that affect bodily temperature [92], or in the case where ambient temperatures trigger affective states [17].

The results of Experiment 1 showed that there was a positive relationship between temperature concepts and the arousal dimension of the emotions. In line with our hypotheses, the results also revealed an inverted U-shaped relationship between temperature concepts and valence since the extreme temperature concepts, both cold and hot, were associated with negatively valanced emotions, whereas the milder ambient temperature was associated with positive valanced emotions. A possibility is that valence is related to embodied process of comfort as warmer temperatures are comfortable but extreme temperatures at both ends can generate discomfort [47]. These results are consistent with Wilkowski et al. [93] as the authors suggested people from different cultures use metaphorical expressions of hot and negative emotions (e.g., anger). These results also are also in line with Baylis et al. [62], who found that expressions of positive emotions in social media were the highest at 20°C and decreased beyond 30°C, at which point negative emotions also increased. It is worth noting that the present study did not control for whether participants interpreted the temperature concepts presented as coming from the environment or from a specific object, despite the visual representations used. Hence the temperature range considered comfortable might differ. The results of Experiment 2 were partially consistent with studies that have implied that warmer temperatures are positively valanced [17, 59, 60]. It is important to note that only two temperature words were used. Experiment 2 also revealed that the association between temperature and arousal was more robust than that between temperature and valence, potentially because associations with valence at higher temperatures is less clear as hotter temperatures can be evaluated positively or negatively, as Experiment 1 showed.

Furthermore, people may associate high arousal emotions, whether they are positively or negatively valanced, with higher temperatures because body temperature increases when they experience those emotions. Some studies that have shown that the temperature of peripheral body regions decreases during negative-valanced, high-arousal emotional states [31, 67, 94, 95]. Nevertheless, other studies have indicated that body temperature changes which are triggered by emotions generally accompany arousal, but are independent of the valence of the emotions [92, 96], which seems to be consistent with the associations of the higher temperature concepts in Experiment 1 and the smaller difference across dimensions of Experiment 2. Examining the inverse relationship between temperature and emotions, in which certain temperatures trigger specific emotional states, the associations can come from high ambient temperatures or activities that increase body temperature and hence arousal. For example,

physical exercise increases body temperature and at the same time may increase excitement and energy levels. Similarly, it is possible that the associations between positive emotions with low levels of arousal and ambient temperatures arise because at this temperature, people are at their homeostatic optimum [1, 12, 30] and therefore feel calm, secure, or happy.

The results of Experiment 1 showed that the associations across the four languages exhibited a high degree of similarity and followed the same overall direction towards the two dimensions of the emotions. These findings are consistent with other studies that have found large similarities in associations between emotions and colors [49, 82, 97, 98] and emotions and brightness [99] across languages. The large similarity in temperature-emotion associations can be the result of highly comparable concepts linked to emotions across languages, which can potentially be captured by broad categories. As Ogarkova [37] suggested, emotional categories in most languages have similar hierarchical structures and the variance of emotion lexicons can be explained by a few relevant dimensions. Another potential explanation of these results is that the subjective experience of emotions did not differ significantly across speakers of the various languages. It is possible that the emotion-temperature associations are fundamentally driven by core affect, which according to the constructionist theory of emotion, is parsed into specific emotion categories [24, 100]. As Sievers et al. [101] suggested, there is a high degree of similarity in how people understand expressions of emotional arousal since they are signaled with a multisensory code based on variations in magnitude. Our findings agree with Jackson et al. [27] in that they seem to reflect the existence of a common semantic framework of emotions across language based on valence and arousal, which are linked to neurophysiological systems that keep homeostasis, although there exists cultural differences.

Despite the high degree of similarity in the emotion-temperature associations across languages, small differences were present. These differences may arise because of linguistic discrepancies and what the various emotions mean in across languages, as well as countries [27]. As Lindquist [36] suggested, languages encode emotions differently, and emotional perception is culturally relative. Additionally, these differences may be caused by environmental factors and the degree of exposure native speakers of a given language that predominantly live-in specific countries have with different temperature ranges. For example, Jonauskaite et al. [49] found that the association between yellow and the concept of joy varied depending on overall exposure to sunshine. Temperature may affect the expression of affect, as well as the subjective experience of similarly intense affective stimuli [12].

Regarding the lower temperature associations for high-arousal emotions in Chinese-speaking participants, it is possible that these differences are the result of a restrained view of the experience and reporting of intense emotions [102]. Intriguingly, there was a slightly higher correlation in the associations between Chinese- and Spanish-speaking participants compared to that between Chinese- and Japanese-speaking participants, as based on geographical and linguistic distance, the latter should be higher [103]. It is possible that this was caused by a greater international cultural exposure from both language groups. However, further research is needed to strip out the effect behind these differences.

## Limitations and future directions

One of the main limitations of the present work relates to the set of emotions used. While we focused on emotions that derived from the valence and arousal dimensions, the pool of emotions that can be studied is virtually endless, and other emotions that could have associations with temperature were not included. For instance, romantic or sensual emotions were not analyzed. Future studies may focus on associations with a much more precise set of emotions that have greater relevance for specific fields or applications. That said, the emotion adjectives [70]

have been validated across cultures in 23 consumer studies (each with 104–270 participants) involving New Zealand and Chinese consumers. The adjectives were also validated with different types of stimuli (i.e., text, images, aromas, and taste). The emotion circumplex covers a wide range of relevant emotions while remaining parsimonious and is applicable to extensive classes of stimuli. Another aspect to consider when applying these temperature-emotion associations in real world scenarios, is that both temperature and emotions can be product- or context-specific. For instance, while companies may want to generate associations between refreshing beverages and positive emotions, using warm temperatures associations would not be ideal.

Another limitation comes from the method in which the temperatures were presented (visual representations in Experiment 1 and temperature words in Experiment 2). Since no actual temperatures were used, it is not possible to rule out potential semantic effects. People could have also interpreted emotion or temperature words differently, thus introducing some variability. In Experiment 1, people from different countries may not be equally used to certain temperatures. in Experiment 2, people could have had considered diverse temperature ranges for the words hot and cold. Nevertheless, the results provide considerable confidence since the experiment captured relative differences given its within-subjects design. The five temperatures and their visual representations (along with the specific values in ˚C and ˚F) in Experiment 1 were chosen as way to cover a broad range of the ambient temperature spectrum, reduce potential language biases, and increase familiarity with temperature measurements. However, it is not certain that participants thought about ambient temperature with these representations. Future studies could expand the range of temperatures and represent them in different ways so that the meaning of temperature is less ambiguous. Moreover, exploring potential differences in the associations between emotions and environmental and object-based temperatures could generate interesting insights. For instance, similar versions of IATs could be designed using pictures of objects or scenes evoking different temperatures combined with facial expressions, such as emojis. Another limitation, especially in the IATs, comes from the possibility that, when evaluating the emotion-temperature associations explicitly or pressing a key in the IAT, participants may not have read the entirety of the pairs of emotion words but instead relied only on the first word. That being said, the use of these emotion adjectives has been extensively validated in multiple studies [70–72].

In recent years, the interest in crossmodal correspondences has seen a rapid growth from academics and practitioners. Research on these correspondences has found a myriad of associations between different modalities (see [104]), and temperature-based correspondences has recently regained the interest of researchers relates to temperature [105, 106]. Spence [107] has recently reviewed the literature on temperature-related crossmodal correspondences. The present study provides valuable insights to advance the study of crossmodal correspondences since the explicit and implicit associations found here may help deepen the understanding of temperature-based crossmodal correspondences mediated by emotions and the role language might play in them. More specifically, these results can guide future studies on the mechanisms behind temperature-based crossmodal correspondences.

To conclude, our findings provide evidence of the existence of consistent associations between emotions and temperature concepts at the explicit level across languages. The findings also provide evidence that some explicit associations also translate to the implicit level. The present study also adds to the literature on emotions and their associations with abstract concepts, and to research on the bidirectionally causal embodied processes between emotions and temperature. Furthermore, the present article contributes to the discussion of how conceptual metaphors can help people understand abstract concepts by interpreting them in terms of

concrete experiences, and how using these metaphors can change both how people view the world and their subsequent behavior.

## Supporting information

**S1 File.**
(PDF)

## Author Contributions

**Conceptualization:** Francisco Barbosa Escobar, Carlos Velasco, Kosuke Motoki, Qian Janice Wang.

**Formal analysis:** Francisco Barbosa Escobar, Carlos Velasco, Qian Janice Wang.

**Investigation:** Francisco Barbosa Escobar, Carlos Velasco, Kosuke Motoki, Qian Janice Wang.

**Methodology:** Francisco Barbosa Escobar, Carlos Velasco, Kosuke Motoki, Qian Janice Wang.

**Project administration:** Derek Victor Byrne.

**Resources:** Derek Victor Byrne.

**Supervision:** Carlos Velasco, Derek Victor Byrne, Qian Janice Wang.

**Visualization:** Francisco Barbosa Escobar, Carlos Velasco, Kosuke Motoki, Qian Janice Wang.

**Writing – original draft:** Francisco Barbosa Escobar, Carlos Velasco, Kosuke Motoki, Qian Janice Wang.

**Writing – review & editing:** Francisco Barbosa Escobar, Carlos Velasco, Kosuke Motoki, Derek Victor Byrne, Qian Janice Wang.

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
