## [Decision Letter · Decision Letter 0]

26 Mar 2021

PONE-D-20-33913

The temperature of emotions

PLOS ONE

Dear Dr. Barbosa Escobar,

Thank you for submitting your manuscript to PLOS ONE. After careful consideration, we feel that it has merit but does not fully meet PLOS ONE’s publication criteria as it currently stands. Therefore, we invite you to submit a revised version of the manuscript that addresses the points raised during the review process.

I was fortunate to review reviews from two researchers in the field who had helpful suggestions and comments for your manuscript, which should guide your revision, which we are looking forward to. 

We look forward to receiving your revised manuscript.

Kind regards,

Hedwig Eisenbarth

Academic Editor

PLOS ONE

Journal Requirements:

"The present work only involved online questionnaires and online implicit association tests. The study complies with the policies and requirements stated by the Aarhus University Research Ethics Committee."

a. Please amend your current ethics statement to confirm that your named institutional review board or ethics committee specifically approved this study.

3. Please provide additional details regarding participant consent.

In the ethics statement in the Methods and online submission information, please ensure that you have specified what type you obtained (for instance, written or verbal, and if verbal, how it was documented and witnessed).

If your study included minors, state whether you obtained consent from parents or guardians.

If the need for consent was waived by the ethics committee, please include this information.

Reviewers' comments:

Reviewer's Responses to Questions

**Comments to the Author**

1. Is the manuscript technically sound, and do the data support the conclusions?

Reviewer #1: Partly

Reviewer #2: Yes

2. Has the statistical analysis been performed appropriately and rigorously? 

Reviewer #1: No

Reviewer #2: Yes

3. Have the authors made all data underlying the findings in their manuscript fully available?

Reviewer #1: Yes

Reviewer #2: Yes

4. Is the manuscript presented in an intelligible fashion and written in standard English?

Reviewer #1: Yes

Reviewer #2: Yes

5. Review Comments to the Author

Reviewer #1: The authors report on 2 studies investigating implicit and explicit associations between temperatures and emotion adjectives. The authors find that cold temperatures are associated with low arousal negative emotions and hot temperatures are associated with high arousal positive emotions (though slightly also with high arousal negative emotions in Study 1). Results are largely consistent across first languages.

These are interesting studies in fairly well-powered samples that advance our understanding of emotion-temperature associations. Overall I think this paper makes a nice next step in the literature. That said, I have some concerns about the analytic approaches and the way the results are integrated with prior literature. I outline these below.

Major concerns

- Background literature that is closely related to the current study is not clearly described. The reader should get a few more details on citations 9-16, the Bergman et al. citation [58], and Bruno et al. [1] (i.e., what do you mean by “emotionally warm.”). Because the methods and directions of the results of these papers are not very clearly described, the reader is left unsure of what exactly the prior literature tells us and what is novel about the current study. I came to understand this in time in the paper, but it should be described more clearly earlier on in the introduction.

- I found the paper’s current reviews of the embodied cognition and evolutionary psych literatures either incomplete or confusing. There’s much more work than just Inagaki’s research on embodied processes, and the evolutionary psych paragraph just references a MRI study that I don’t think actually supports the authors’ argument… I actually feel like much of this paper’s results could more elegantly and succinctly summarized by just grounding it in the constructionist theory of emotion (see reviews by Barrett, 2017, 2006, as well as relevant papers by Lindquist, Satpute, etc.). This framework suggests that emotion concepts are used to parse core affect into specific emotion types, and it makes sense that temperature would feed into emotion concepts. Sievers et al. also have a recent paper in PNAS that could help with this multimodal representation of affect/emotion.

- At times the authors make claims like “emotions have meanings and are experienced differently based on the concepts with which they are associated” and “there is a high degree of universality in the experience of emotions,” but these claims are a huge topic of debate. The first is a tenet of constructionist theory and the second is a claim of the basic emotion theory. I’d either remove or phrase claims like these more speculatively. It also seems unclear what the authors’ stance is when these opposing claims are both held in one paper.

- One methodological concern is that recruiting through professional networks could introduce bias? Was this controlled for? What were the Ns for different recruitment types?

- Similarly, is there any assurance that different language groups are not confounded by other variables? The age differences are pronounced. Could you test for significant differences across these? Do results vary when this is controlled for?

- I found it odd that the study design involved presenting PAIRS of emotion words. How do we know that (some) readers aren’t just reading the leftmost word (esp in study 2 when speed is critical)? This is a methodological detail that should be highlighted earlier on for readers and noted as a limitation. I think it makes interpretation less clear cut..

- I’m not sure I follow the authors’ PCA analysis. Specifically, I think it’s odd to add in valence/arousal values, especially when these values are essentially made up by the authors. Are there any normative data that they could use instead? Or instead, I would just encourage the authors to run the PCA on the temperature analyses and then show that the two axes essentially line up with valence and arousal. Right now, they make interpretations of how “valence was positively related to ambient temperature (20 °C)” and that “valence was related to ambient or comfortable temperature concepts and arousal was related to higher temperatures,” but I don’t think these interpretations are really logical conclusions from PCA… The authors essentially just made up the valence/arousal values, and factors are only associated with certain variables after controlling for other factors… I like the idea of applying PCA, but I don’t think the way they’re executing/interpreting it is very strong.

- In fact, my interpretation of the Fig 4 plot is that there is nice evidence that different temperatures are essentially associated with different areas of the circumplex (the “peak” of the wave of emotion associations moves from far right to the middle then to the left of each plot as temperature increases, suggesting that people are essentially moving counterclockwise through the circumplex starting in the low left and ending in the upper middle as temp increases). Is there some way to rethink the analytic plan so that temperature is treated as a continuous variable, rather than a series of independent measures? I can’t think of a way right now, but perhaps consulting with a statistician could reveal some kind of interesting multidimensional scaling approach?

- I also feel like greater statistical expertise might be needed to think through the language analysis… Right now, it seems like the authors correlate matrices that are just average ratings across all speakers of each language… This sucks out all variance within each language, a suboptimal approach. Is there some way to do a nested analysis? Perhaps not, but thinking this through would be helpful.

- I’m glad the authors point out the important limitation that people may have thought of “hot” and “cold” as referring to affective states, not temperatures. Perhaps a future direction would be to use images or phrases associated with hot things in the IAT (e.g., press left if you see a picture of a warm scene/activity or a picture of a positive facial expression. Press right if you see a picture of a cold scene/activity or a negative facial expression).

Smaller concerns

- I’m not sure about the phrase “temperature concepts.” Do humans really have a specific concept around “20 degrees Celsius?”

- I found myself tripping up with interpretation phrases like “participants associated the emotions Enthusiastic/Inspired and Passive/Quiet with 20°C.” It seems like the authors are basing these on the ANOVA results showing that average ratings for the 20°C question are higher for these emotions than others. But just because these ratings are higher for these emotions than others doesn’t really mean that they “significantly associate” these concepts… I feel like two adjustments are needed here: 1) when talking about average ratings in the task just refer to them as “average ratings were higher for X emotion than Y emotion” rather than saying that there were “significant associations” and 2) I might stick to just talking about how temperatures showed “peaks” around certain parts of the circumplex rather than overinterpreting which emotions were higher than all the others.

- In abstract, please specify the Ns of study and clarify which language is studied in Study 2

- In intro, maybe use the phrase “chilled with fear” instead of “cold fear”? I don’t know if I’ve ever heard the second in common parlance.

- How exactly does “this set of emotion words mitigate the risk of vague emotions and poor usability?”

- Could Fig 2 include the average ratings in each cell of the table?

- I think the phrase “on the other hand” is misused in the paper. It is used as a phrase for “additionally” when I think it usually implies “for an opposing perspective.”

- I couldn’t follow the authors’ evolutionary argument about blushing. Why is greater heat loss in these areas adaptive? How this gets at the conceptual overlap between temperature and emotion that the authors are studying?

Reviewer #2: Dear Mr. Escobar,

I have had the opportunity to review the authors’ submission, “Temperature of Emotions”. This paper investigates whether temperatures are systematically and specifically associated with emotions. The authors conduct two studies to assess this relationship: In study 1, individuals in four cultures, evaluated 12 emotion terms (sampled from around the circumplex) on five temperatures (ranging every 10 degrees C from 0 to 40). In study 2, an adapted version of the IAT was used to see whether (participants in English) showed implicit biases to happy (+ valence)/active (high arousal) and warm pairings, and unhappy (- valence)/passive (low arousal) and cold pairings.

Overall, this paper is extremely well-written, with a solid introduction and review of the extant literature, as well as sound in design and analyses. The results are straight-forward, aligned with the authors’ hypotheses, and the implications for importance are well-stated. I was also impressed by the detail to design (counterbalancing in both studies, use of the SAM in study 1, and the statistical presentation (CIs, comparison of effects sizes). Figures were nicely displayed and all statistics conformed to the latest recommended APA standards and were thoughtfully-conducted.

The only criticism I have is one already noted by the authors. That is, the collection of reaction time on individual devices, which vary in speed, etc. Although the study is within-subjects (helping to reduce differences among platforms/computer speeds), I still find it concerning to rely on differences of < 200 ms as statistically different when reported from uncontrolled/standardized devices. There is not much to be done, at this point, but perhaps a replication in the lab is possible in the near future (I am assuming in person data collection was limited as a result of the COVID pandemic).

I find the overall subject to be of interest and timely, and of appropriate content for this journal. It not usual that I have few to no concerns of criticisms before publication. As it stands, I only have a few clarifying points.

Minor Criticisms:

1. It would be interesting to know in Study 1 the location (country) from which participants originated or currently-lived. This might help disentangle cultural from linguistic effects. Relatedly, in Study 2, it would be nice to know whether English-speaking participants (all living in the UK) were familiar with another language.

2. I am not sure how ATR statistics (used through R) are similar/different to the Wilcoxon non-parametric signed-based statistics. It would be helpful to make a comparison to these more “common” statistics. There seems to be some debate as to whether homogeneity of variances is required for the Wilcoxon (which you say you do not have).

3. Although the results for Study 1 are clear and presented both across languages/cultures and for each separately, I would like to see some discussion (even if speculative) as to why Chinese and Spanish showed higher correlation than presumably between Chinese and Japanese (because it is not listed as higher or lower). To that end, it would also be interesting to expound on why there are specific some cultural/linguistic differences (albeit minimal to overwhelming consistent pattern across languages).

4. I would appreciate a further discussion of the improved scoring methods of the d statistic of Greenwald and colleagues. I do not follow the use of replacing the error trials with the mean RT + 2 SDs.

Overall, I believe that the research is potentially interesting and worthy of publication.

Thank you for the time to review this paper.

Kind Regards,

6. PLOS authors have the option to publish the peer review history of their article (what does this mean?). If published, this will include your full peer review and any attached files.

Reviewer #1: No

Reviewer #2: **Yes: **Jennifer M B Fugate

---

## [Author Response · Author response to Decision Letter 0]

22 Apr 2021

Dear Dr. Eisenbarth,

We thank you for your time and the oppotunity to revise and improve our manuscript. Attached, you will find the response to reviewers and the manuscript with and without track changes in PDF format, as well as their LaTeX source files.

Best wishes,

Francisco Barbosa Escobar, on behalf of all authors.

---

## [Decision Letter · Decision Letter 1]

7 May 2021

PONE-D-20-33913R1

The temperature of emotions

PLOS ONE

Dear Dr. Barbosa Escobar,

Thank you for submitting your manuscript to PLOS ONE. After careful consideration, we feel that it has merit but does not fully meet PLOS ONE’s publication criteria as it currently stands. Therefore, we invite you to submit a revised version of the manuscript that addresses the points raised during the review process.

Thank you for submitting your revised version, it addresses the points of the reviewer very well. Both reviewers just had some minor revision suggestions, which should be easy to address so I invite you to make those revisions.

We look forward to receiving your revised manuscript.

Kind regards,

Hedwig Eisenbarth

Academic Editor

PLOS ONE

Journal Requirements:

Additional Editor Comments (if provided):

Thank you for submitting your revised version, it addresses the points of the reviewer very well. Both reviewers just had some minor revision suggestions, which should be easy to address so I invite you to make those revisions.

Reviewers' comments:

Reviewer's Responses to Questions

**Comments to the Author**

1. If the authors have adequately addressed your comments raised in a previous round of review and you feel that this manuscript is now acceptable for publication, you may indicate that here to bypass the “Comments to the Author” section, enter your conflict of interest statement in the “Confidential to Editor” section, and submit your "Accept" recommendation.

Reviewer #1: (No Response)

Reviewer #2: All comments have been addressed

2. Is the manuscript technically sound, and do the data support the conclusions?

Reviewer #1: Yes

Reviewer #2: Yes

3. Has the statistical analysis been performed appropriately and rigorously? 

Reviewer #1: Yes

Reviewer #2: Yes

4. Have the authors made all data underlying the findings in their manuscript fully available?

Reviewer #1: Yes

Reviewer #2: Yes

5. Is the manuscript presented in an intelligible fashion and written in standard English?

Reviewer #1: Yes

Reviewer #2: Yes

6. Review Comments to the Author

Reviewer #1: The authors have done a great job incorporating edits and the manuscript is much improved. I just have a few small points on the revision that I think will improve interpretability.

- PCA results are interesting. Dim 1 seems to be arousal and Dim 2 is valence. This is opposite most findings (Dim1 = valence, Dim 2 = arousal), but it sometimes happens depending on the stimuli included. But I think it needs to be narrated for readers a bit. Perhaps labeling the axes, adding notes in the main text, and even labeling the quadrants would be helpful. I also don’t think you need to interpret 20 degrees “aligning” w the vertical axis and 30 degrees “aligning” w horizontal axis (alignments are weak and not quite sure how to interpret them…). Finally, the rotation is described as counterclockwise around the circumplex, but doesn’t increasing temp move clockwise on the PCA projection? In the discussion I agree that it moves counterclockwise in a “canonical” circumplex, but the differences between what is found canonically and what is presented in the figure are never clarified for readers.

- The LMM analysis is v important and interesting. I think the results could be more fully discussed. In particular, there are some interesting differences between languages that are not explored at all. There’s a few sentences about it in the general discussion, but talking through a few of the findings in the results of Study 1 seems merited.

Reviewer #2: Dear Dr. Escobar,

Thank you for your revision on the ms: “The temperature of emotions”. I found that it addressed my initial concerns and those brought up by the other reviewers. I found myself actually agreeing with many of the initial points of the other reviewers, despite my very positive initial review. To that end, you have not only acknowledged and sufficiently addressed my concerns, but you have also addressed their concerns (to my satisfaction, although of course they may disagree).

The one point that I did not mention initially, and would like to strongly recommend you elaborate upon even a bit more, is the theoretical grounding of Psychological Constructionism. In fact, I have published on this work extensively, and have a color-emotion paper that uses the Theory of Constructed Emotions (Barrett, 2006, 2019) to explicitly test whether colors and emotions pairings are both specific and consistent (as that theory would expect). I believe that you would find that paper informative and perhaps some of the language within the rationale might even help you frame your comment to reviewer 1, comment 2.

Beyond that, I am happy and satisfied with the revisions. I will recommend to the editor that your submission be published.

Kind Regards,

Jennifer Fugate

________

References:

Fugate, J. M. B., & Franco, C. L. (2019). What color is your anger? Assessing color-emotion pairings in English speakers. Frontiers in psychology, 10, 206.

7. PLOS authors have the option to publish the peer review history of their article (what does this mean?). If published, this will include your full peer review and any attached files.

Reviewer #1: No

Reviewer #2: **Yes: **Jennifer M.B. Fugate

---

## [Author Response · Author response to Decision Letter 1]

11 May 2021

Dear Dr. Eisenbarth,

We are grateful for your time and the opportunity to revise and improve our manuscript.

Attached, you will find the response to reviewers and the manuscript with and without

track changes in PDF format, as well as their LaTeX source files.

Best wishes,

Francisco Barbosa Escobar, on behalf of all authors.

---

## [Decision Letter · Decision Letter 2]

17 May 2021

The temperature of emotions

PONE-D-20-33913R2

Dear Dr. Barbosa Escobar,

We’re pleased to inform you that your manuscript has been judged scientifically suitable for publication and will be formally accepted for publication once it meets all outstanding technical requirements.

Kind regards,

Hedwig Eisenbarth

Academic Editor

PLOS ONE

Additional Editor Comments (optional):

Reviewers' comments:

Reviewer's Responses to Questions

**Comments to the Author**

1. If the authors have adequately addressed your comments raised in a previous round of review and you feel that this manuscript is now acceptable for publication, you may indicate that here to bypass the “Comments to the Author” section, enter your conflict of interest statement in the “Confidential to Editor” section, and submit your "Accept" recommendation.

Reviewer #1: All comments have been addressed

2. Is the manuscript technically sound, and do the data support the conclusions?

Reviewer #1: Yes

3. Has the statistical analysis been performed appropriately and rigorously? 

Reviewer #1: Yes

4. Have the authors made all data underlying the findings in their manuscript fully available?

Reviewer #1: Yes

5. Is the manuscript presented in an intelligible fashion and written in standard English?

Reviewer #1: Yes

6. Review Comments to the Author

Reviewer #1: The authors have addressed all my concerns and have produced a nice contribution to the field. Congrats!

7. PLOS authors have the option to publish the peer review history of their article (what does this mean?). If published, this will include your full peer review and any attached files.

Reviewer #1: No

---

## [Editor Report · Acceptance letter]

24 May 2021

PONE-D-20-33913R2 

The temperature of emotions 

Dear Dr. Barbosa Escobar:

I'm pleased to inform you that your manuscript has been deemed suitable for publication in PLOS ONE. Congratulations! Your manuscript is now with our production department. 

Kind regards, 

on behalf of

Dr. Hedwig Eisenbarth 

Academic Editor

PLOS ONE